# High Acceptance of COVID-19 Tracing Technologies in Taiwan: A Nationally Representative Survey Analysis

**DOI:** 10.3390/ijerph19063323

**Published:** 2022-03-11

**Authors:** Paul M. Garrett, Yu-Wen Wang, Joshua P. White, Yoshihsa Kashima, Simon Dennis, Cheng-Ta Yang

**Affiliations:** 1School of Psychological Sciences, University of Melbourne, Melbourne 3010, Australia; paul.garrett@unimelb.edu.au (P.M.G.); josh.white@unimelb.edu.au (J.P.W.); ykashima@unimelb.edu.au (Y.K.); simon.dennis@gmail.com (S.D.); 2Department of Psychology, National Cheng Kung University, Tainan 701, Taiwan; r06227122@g.ntu.edu.tw; 3Unforgettable Research Services, Melbourne 3010, Australia; 4Graduate Institute of Mind, Brain and Consciousness, Taipei Medical University, Taipei 110, Taiwan

**Keywords:** COVID-19, tracking technologies, SARS-CoV-2, contact tracing, Taiwan, public health, health policy, privacy, privacy calculus

## Abstract

Taiwan has been a world leader in controlling the spread of SARS-CoV-2 during the COVID-19 pandemic. Recently, the Taiwan Government launched its COVID-19 tracing app, ‘Taiwan Social Distancing App’; however, the effectiveness of this tracing app depends on its acceptance and uptake among the general population. We measured the acceptance of three hypothetical tracing technologies (telecommunication network tracing, a government app, and the Apple and Google Bluetooth exposure notification system) in four nationally representative Taiwanese samples. Using Bayesian methods, we found a high acceptance of all three tracking technologies, with acceptance increasing with the inclusion of additional privacy measures. Modeling revealed that acceptance increased with the perceived technology benefits, trust in the providers’ intent, data security and privacy measures, the level of ongoing control, and one’s level of education. Acceptance decreased with data sensitivity perceptions and a perceived low policy compliance by others among the general public. We consider the policy implications of these results for Taiwan during the COVID-19 pandemic and in the future.

## 1. Introduction

Taiwan’s response to the COVID-19 pandemic, which was caused by the severe acute respiratory syndrome coronavirus 2 (SARS-CoV-2) virus, is considered an international ‘Gold Standard’ in infectious disease control [1]. Rapid border closures, increased mask manufacturing, and immediate home quarantines for international arrivals monitored by law enforcement officials were introduced by Taiwan’s Central Epidemic Command Centre to stop the spread of the virus [2]. These policy decisions, combined with social behaviors, such as social distancing, hand washing, and mask wearing [3], have proven effective at preventing the spread of SARS-CoV-2, with fewer than 1000 total cases recorded between January 2020 and March 2021 [4]. However, the asymptomatic and highly infectious nature of SARS-CoV-2 makes an outbreak only a matter of time, as exemplified by the May 2021 Taiwan outbreak, where daily cases rose from 5 to 723 in less than two weeks [4]. Constant vigilance is necessary to prevent future outbreaks.

To this end, the Taiwanese Government introduced the ‘Taiwan Social Distancing App’ [5]. This app uses the Apple/Google Bluetooth exposure notification (EN) system [6] to anonymously register nearby phones. However, for this app to be effective, it must receive wide public support, and the perceived health benefits of the app must outweigh any perceived risks and harms associated with the tracing app [7].

### 1.1. Mobile Tracing Technologies

Mobile tracing technologies use a person’s smartphone to alert nearby mobile carriers if they have had contact with someone who is infected with COVID-19 [8]. In the case of tracing apps, a registry of nearby devices is recorded on the user’s phone and provides alerts if the user becomes infected. By contrast, telecommunication systems notify individuals en masse if someone becomes infected within a designated area. Importantly, these alerts do not require the user to consent to receiving these notifications. In both instances, the choice to identify oneself as infected through a tracing technology is typically left to the individual. Whether someone chooses to use a tracing technology may depend on their perceived risks and benefits. Figure 1 (top) displays the benefits and risks associated with three common mobile tracing technologies: telecommunication tracing, GPS tracing, and Bluetooth tracing, which we detail below.

Telecommunication tracing has already been used as part of Taiwan’s ‘electronic fence’ [9] to monitor individuals in home quarantine, and is effective at monitoring large movements (e.g., driving to work [10]) and alerting individuals en masse when an outbreak occurs. However, it has poor spatial resolution, making it ineffective at alerting individual close contacts who are likely to have contracted the virus. GPS tracing is effective at targeting specific locations [11]; however, it performs poorly indoors or in built-up areas, making it an unreliable contact-tracing technology. Bluetooth tracing apps function within a limited radius, are obscured by objects such as walls, and register anonymous device identifiers [12,13]. For these reasons, Bluetooth COVID-19 tracing apps are favored by both international governments and private developers (e.g., the COVIDSafe app [14], Germany’s CORONA-Warn App [15], the exposure notification system [6], and Taiwan’s ‘Social Distancing App’ [5]).

Once a mobile device records nearby users, these user registries may be stored in either a centralized or decentralized fashion [16]. Centralized systems store information about a user’s contact history on a government-controlled server accessible by health officials, which can be used to supplement manual tracing efforts. Decentralized systems store user’s contact registries on a personal phone. If an individual identifies as infected, they alert their registered contacts directly, phone-to-phone. Under both systems, exposure alerts require the user to test positive with the virus, and to enter a verification code provided by a health official to avoid false alarms.

Taiwan’s new ‘Social Distancing App’ uses the Apple/Google EN system to register nearby devices and stores data in a decentralized fashion [5]. The Government is responsible for the app and individuals are notified if they have been within two meters of an infected individual for more than two minutes over the past 14 days. Data are deleted from the phone after 28 days, and if a contact registry is uploaded to notify users, then these data are deleted from the server after 10 days.

In the current study, we assessed the national acceptance of three tracing scenarios—a decentralized Bluetooth app using the EN system (very similar to Taiwan’s ‘Social Distancing App’), a centralized government Bluetooth app, and a telecommunication tracing system (similar to Taiwan’s electronic fence; see bottom panel, Figure 1).

### 1.2. Technology Acceptance

As the success of tracing technologies is dependent upon their perception of acceptability and uptake within the community (particularly given that they generally require the user’s consent), it is imperative for governments and policy makers to consider how to maximize their acceptance to achieve the most positive public health benefits [17]. The health belief model [18] identifies six critical factors for the success of an effective health policy: policy benefits and barriers, perceived illness susceptibility and severity, self-efficacy, and cues to action. Determining which of these factors predict the acceptance and uptake of tracing technologies, such as Taiwan’s ‘Social Distancing App’, is an important public health issue and may prove less intuitive than one first assumes.

On one hand, a COVID-19 tracing app provides a potential benefit to public health—faster contact tracing and a faster return to unrestricted social mobility when outbreaks occur [19]. However, tracing apps may also pose a risk to one’s personal data privacy and security [12,20,21], creating a barrier to app uptake. This weighing of personal costs against public health benefits has been termed ‘privacy calculus’ [7,22] and may be influenced by other transient factors, such as illness susceptibility and severity (see [23] for a wider discussion of other factors).

For example, while cases are low, people’s internal privacy calculus may place little weight on the susceptibility or risk of infection for themselves and those around them. This in turn may sway them to not download an app on the grounds of personal privacy and security. However, these internal metrics may change with an increase in cases or deaths caused by the virus (i.e., perceived illness severity), as well as impacts such as a loss of employment, or by discovering that someone close to oneself has become infected. These transient impacts must be accounted for when considering the public uptake of a tracing app.

Similarly, one’s psychological well-being may play a role in deciding to use a tracing app. Recent research suggests psychological resilience—how effectively people deal with the adversity of stressful events, tragedy, and trauma—is associated with less concern about the pandemic and a lower level of generalized anxiety and depression [24], which may influence whether tracing apps are perceived to be necessary or even beneficial to public health. However, a lower concern about the pandemic may lead people to seek social interactions more frequently, making a tracing app more beneficial to these individuals. It is important to allow these individuals to engage with a tracing app when they need to provide a means of self-efficacy and proactive engagement with the app and health policy.

Finally, one’s attitude towards their government and their neoliberal worldviews on how the Government should interface with the lives of their citizens, may affect the acceptance and use of COVID-19 mobile tracing technologies [25]. In this study, we aim to use the framework of the health belief model [18] to determine how COVID-19 perceptions and impacts, psychological resilience, and governmental and neoliberal worldviews influence the privacy calculus governing the acceptance of COVID-19 tracing technologies in Taiwan.

Although acceptance of a mobile tracing app does not necessarily translate to public uptake (see, [23,26,27]), determining public support is an important step towards determining which infrastructure and technologies countries should invest in to combat the spread of COVID-19 and similar viruses [25,28].

To this end, the current study investigated the conditions under which three mobile tracing technologies (Figure 1) would be deemed acceptable in a nationally representative sample of the Taiwanese population. This study extends upon international work conducted in Australia [23], the United Kingdom [25], Germany [27], and on young adults in Taiwan [28], that previously provided descriptive accounts of the COVID-19 tracing app acceptance. Here, we apply Bayesian analysis tools to determine predictive factors for App acceptance in a representative sample of Taiwan, using survey items constructed within an established health policy framework—the health belief model.

## 2. Materials and Methods

### 2.1. Participants

Four nationally representative samples of 1500 participants (6000 in total, 2000 per scenario) completed a 15 min online survey. Representative sampling was completed through the survey distribution company ‘Gosurvey’ at a cost of USD 21,500, and participants were reimbursed per their agreement with Gosurvey. Surveys were conducted at one-week intervals, starting 8 April 2020 and ending 29 April 2020. Only participants who passed a comprehension check indicating that they had read and understood the contact tracing technology (scenario) presented to them were included in the main analyses of this paper. Many participants did not pass the comprehension check and were excluded, possibly due to an English-to-Chinese translation issue or miscommunication error. To ensure our findings were not adversely affected, modeling was completed with and without participant exclusions and no meaningful differences were observed (see Appendix A). Demographics are displayed in Table 1.

### 2.2. Design and Procedure

Figure 2A displays the survey design and procedure, and Figure 2B displays key survey items. Participants provided informed consent before being assessed for their psychological resilience using the 25-item Connor-Davidson Resilience (CD-RISC) Scale (the CD-RISC scale items add up to between 0 and 100, with higher values indicating higher resilience, and has previously displayed a high degree of internal consistency: Chronbach’s alpha = 0.89, test–retest reliability, and Person’s r = 0.87 [29]).

Resilience was followed by items assessing participant’s demographics, COVID-19 perceptions and impacts, and government perceptions, before reading one of three hypothetical COVID-19 mobile tracing scenarios (telecommunication tracing, a centralized Government app, or the Bluetooth exposure notification system).

Participants then answered questions about the perceived benefits, trust, and potential harms associated with the technology, before indicating if they would ‘accept’ the introduction of this technology? ‘No’ responses were followed by items assessing if attitudes would change (i) if data were stored only for six months (‘Sunset clause’), (ii) if there was an opt-out option (telecommunication scenario), and (iii) if data were stored locally on the users’ phones (Government app scenario). Finally, attitudes towards immunity passports (not assessed in this paper) and neo-liberal worldviews (attitudes reflecting a belief that the free market is best at meeting the societal and financial needs of the population) were assessed, before a participant debrief. The exact questions asked to participants are shown in Table 2.

### 2.3. Scenario Descriptions

A summary of the three contact tracing scenarios presented to participants is produced below. For the full details of the exact scenarios presented to participants, see Appendix A. The telecommunication network tracking scenario described mandatory mobile tracking with no possibility to opt-out. Data would be stored in an encrypted format on a secure server accessible only to the Taiwanese Government, who may use the data to locate people who violated lockdown orders, enforcing these orders with fines and arrests where necessary.

The Government app scenario described a voluntary, centralized COVID-19 tracing App. Data would be stored in an encrypted format on a secure server accessible only to the Taiwanese Government, and would only be used to contact those who might have been exposed to COVID-19.

The Bluetooth scenario described a voluntary, decentralized contact tracing app that would use Bluetooth to help inform people if they had been exposed to others with COVID-19. The Government would not know the identities of these individuals. This scenario was based on Apple and Google’s EN system and closely resembles Taiwan’s current ‘Social Distancing App’.

### 2.4. Data Analysis and Reporting

Participants were excluded from the analysis if they did not pass the comprehension check item (*N* = 2175). The included results with these participants are presented in the Appendix A. All data, code, and model fits for this project are freely available through our repository on the Open Science Framework, https://osf.io/u28n7, accessed on 1 December 2021.

#### 2.4.1. Likert Comparisons

Bayesian ordinal probit regressions were used to directly compare Likert-responses using the MCMCoprobit and HPDinterval functions in R packages MCMCpack [30] and Coda [31], respectively. This method assumes that there are latent normally distributed continuous variables that underlie ordinal responses. These latent variables are then segmented into ordinal Likert responses by C–1 (number of response options–1) thresholds. To set the location of the underlying latent variable and make the model identifiable, the lowest threshold parameter is fixed at zero [32] and all other thresholds are estimated. Data samples were modeled together to ensure consistent threshold parameters across the Likert-items (see [33], for model details), making the items and their 95% highest density intervals directly comparable. This form of Bayesian posterior reporting avoids the need for Bayes Factors, and instead, allows the reader to directly view the credible interval for each variable’s posterior estimate.

#### 2.4.2. Predictive Modeling

Bayesian mixed effects logistic regression modeling was used to predict participants’ acceptance of each COVID-19 contact tracing technology using demographics, perceptions and impacts of COVID-19, COVID-19 cases and deaths, neoliberal and governmental worldviews, and technology attitudes treated as additive functions of technology acceptance. Random intercept effects were included to account for dependencies introduced by each of the three tracing technologies, allowing us to model acceptance across the three scenarios. Likert ratings were treated as numeric data for the purposes of modeling, and all non-categorical variables were grand mean standardized prior to analysis to have a mean of 0 and standard deviation of 1. The four samples were collapsed into a single data set for this analysis because (as the reader will see in the Results section) the four samples were not meaningfully different.

Posterior distributions of model parameters were estimated using Hamiltonian Markov Chain Monte Carlo No-U-turn Sampling implemented in Stan via the R package BRMS [34,35]. Four chains with 4000 iterations and 2000 burn-ins were used. Non-informative priors were set for the intercept and random effect standard deviation parameters (both Cauchy distributions centered on 0 and a scale parameter of 2.5), and fixed effects were estimated from weakly informative priors with a Laplacian distribution centered on 0 and a scale parameter of 1.

## 3. Results

The full results of COVID-19 perceptions, governance and worldview items are included in the Appendix A and are summarized here. COVID-19 perceptions remained constant across samples, with participants on average reporting that they were somewhat concerned for themselves and very concerned about others, and that the virus posed a very severe threat to both themselves and others (Appendix A). Neoliberal worldview items were similarly constant across samples, with participants somewhat agreeing with neo-liberal free-market attitudes (Appendix A). Perceptions of the effectiveness of the Government’s COVID-19 response decreased slightly across the samples from moderate to a bit satisfied (Appendix A), and COVID-19 impact measures (days in lockdown, job loss, and positive cases) were comparable across collection dates (see Table 3). Due to the similarity in the samples, data samples were collapsed and reported as a single sample in the subsequent models.

### 3.1. Tracing Technologies

Figure 3 displays the mean ordinal regression posterior distributions and associated Likert-style responses for items querying people’s perception of the proposed COVID-19 tracing technologies, with items grouped by perceived benefits, trust, and harm. Individual items are described in Table 2. Perceived benefits for each technology were comparable (categorized as ‘a lot’) with telecommunication tracing being perceived as slightly more capable of lowering the risk of infection. Trust items were similarly comparable, ranging from ‘moderate’ to ‘a lot’, with security perceived as the highest for the decentralized Bluetooth app. Harm items differed between technologies—telecommunication tracing was perceived as ‘extremely’ risky, very difficult to decline, collecting the most sensitive data and being the hardest to maintain control of. The Bluetooth and Government apps were comparable in their harm assessments.

### 3.2. Acceptance

Figure 4 shows the acceptability and the conditional acceptability (the acceptability of the technologies given certain conditions are met) of each tracing technology. Conditional acceptability shows the subsequent increase in acceptability under a sunset clause (where data are deleted after six months) and with an opt-out or local storage option. Baseline acceptance was high across all three tracing technologies (67–73%), being highest for Bluetooth and then telecommunication tracing. Acceptance increased with the additional privacy measures, such as a sunset clause (77–82%), local storage option (83%) and opt-out clause (88%).

### 3.3. Regression Modeling

Figure 5 displays the posterior estimates from the Bayesian generalized linear mixed effects model of tracing technology acceptance using demographics, COVID-19 perceptions and impact, government perceptions, technology perceptions, psychological resilience, and worldviews as additive factors, with a random intercept for each hypothetical technology (not displayed in the figure). Error bars display the 95% highest density interval (hereinafter called credible intervals, or CI). The posterior estimates and 95% credible intervals of the random technology (scenario) intercepts are displayed alongside all parameter coefficients in Appendix A. The global intercept had a mean of 0.37 (95% CI −0.81:1.34). Total intercept means (random intercept plus global intercept) for each scenario were ordered from lowest to highest, the Government app (M = 0.16, 95% CI −0.54:0.95), telecommunication tracing (M = 0.43, 95% CI −0.23:1.25), and Bluetooth tracing (M = 0.44, 95% CI −0.22:1.26). To ensure our participant exclusion criteria (our scenario comprehension check) did not alter our findings, we reran this model without excluding participants based on our comprehension check (see Appendix A). Findings were nearly identical to those reported in the main text.

Variables that were predictive of technology acceptance—those where the 95% highest density interval does not cross zero—included education (both university and high school graduates compared to those who did not graduate high school), and many of the technology perception items. The following variables all had a positive predictive relation with tracing technology acceptance: trust in data privacy and security, trust in the provider’s intent, the belief that technology will return one to normal activities sooner and reduce the spread of the virus, and the ability to maintain ongoing control over one’s data. To understand the implications of these results, we report results in terms of how their proportionate increase (or decrease) affects the odds of accepting these hypothetical tracing technologies. Belief that the technology will lower the chance of becoming infected was the most predictive factor, with a 0.57 unit increase in this variable corresponding to a 1.77-fold increase in the odds of accepting the introduction of a tracing technology. Psychological resilience and attitudes towards social distancing overlapped zero (by 1% of their 95% CI), bordering our criteria as positive predictive variables.

Predictive variables against the introduction of tracing technologies included a belief that others were not complying with government policies, and perceptions about the sensitivity of the data being collected. Data sensitivity was the most predictive variable against acceptance, with a 0.22 unit increase corresponding to a 1.25-fold decrease in the probability of accepting a tracing technology. Posterior estimates for small-government interference overlapped zero (by 1% of its 95% CI), bordering our criteria for a negative predictor.

## 4. Discussion

In April 2020, we asked four representative samples of the Taiwanese public to report on their psychological resilience, the perceived risks of and impacts posed by COVID-19, and perceptions of their Government and neoliberal ‘free market’ worldviews, before rating the acceptability of three hypothetical COVID-19 tracing technologies: telecommunication network tracing, a decentralized Bluetooth exposure notification-style system backed by Apple and Google, and a centralized government app. Acceptance was high across all three technologies and improved with additional privacy measures. As per the health belief model, technology acceptance increased with the perceived technology benefits (resuming activities, reducing spread and infection, and data privacy and security), self-efficacy (ongoing data control), and calls to action (indirectly measured by trust in the technology provider). Acceptance decreased with technology barriers (perceived data sensitivity and compliance by others). The remaining factors of the health belief model—virus susceptibility and severity—did not impact our predictive model, possibly reflecting the low case numbers recorded in Taiwan at the time of the survey.

### 4.1. Principal Results

Acceptance of each tracing technology was moderate-to-high overall (67–73%) and improved with additional privacy measures, such as a sunset clause (up to 82%), local storage option (83%), or opt-out option (88%). These results display an overwhelming level of acceptance among the proposed tracing technologies, on par with acceptance levels observed in Western countries, such as Germany (80%) [27] and the United Kingdom (87%) [28], whose data were collected during large-scale outbreaks. As per the health belief model and privacy calculus, we expected acceptance in Taiwan to rise following their first major COVID-19 outbreak (May 2021), wherein the perceived virus susceptibility and severity increased. Indeed, acceptance translated to uptake during this outbreak when the ‘Taiwan Social Distancing App’ became the most downloaded app in Taiwan in both the Apple and Android app stores [36]. Privacy calculus is also apparent in our results with acceptance improving after the inclusion of additional privacy measures, suggesting participants are weighing the costs and benefits of agreeing to use these technologies. This privacy calculus feeds into how technologies are compared, and whether they are accepted or not.

Telecommunication tracing was perceived as potentially the most harmful, while both centralized and decentralized apps were comparable in their harm perceptions. With the perceived benefits and trust items comparable across scenarios, it seems that the Taiwanese public differentiated between potential tracing technologies based on their harm profiles and not on their benefits. However, these public health benefits play an important role in whether individuals are ultimately willing to accept the proposed technology.

Modeling suggests participants’ acceptance ratings were based on the technology’s benefits, privacy, security, control, and trust in the provider’s intent, in addition to the user’s education. Non-acceptance was informed by perceptions of poor public compliance and data sensitivity. Knowing someone who has had COVID-19 appears to play a positive role in acceptability decisions; however, this was limited in our modeling by the low case-numbers at the time of data collection. This agrees with our finding that COVID-19 concerns and perceived virus severity were higher for others than for oneself.

Our results indicate a potential effect of psychological resilience and social distancing perceptions towards the acceptance of tracing technologies. Similarly, we see perceptions shift away from acceptance with the supporting of small-government interference. These posterior estimates bordered on zero and did not pass our arbitrary 95% credible threshold. With national and international attitudes being shaped by the state of the pandemic (e.g., high vs. low case numbers, vaccine availability), these variables may provide potential avenues of further study as the public attitudes and the pandemic evolve. Interestingly, we observed no evidence that the impact of COVID-19 nor the perceptions of government effectiveness (excluding public compliance) influenced technology acceptance.

### 4.2. Limitations

The current study assessed a limited set of tracing technologies within a short window of the pandemic timeline. These limitations impact the generalizability of our results, making it difficult to infer about the acceptance of other potential tracing technologies or how attitudes will shift with the changing nature of the pandemic. Fortunately, our work provides a clear point of reference for national attitudes towards Taiwan’s real-world policy choice, the ‘Social Distancing App’, shortly before its introduction and when COVID-19 case numbers were close to zero. App uptake data have not been released since the app launched; however, inferences from studies on acceptance and uptake in Australia [23] and Germany [27] suggest the uptake to be 18–20% lower than acceptance, making an informed uptake estimate for Taiwan at 53%. This is lower than the 60% necessary for full effectiveness [8,23], but still within a range to assist early isolation efforts. We leave the empirical assessment of this estimate to future studies.

At the time of this study, a tracing app was yet to be released in Taiwan, meaning our hypothetical scenarios lacked a measure of technology effectiveness. Many tracing apps, such as Australia’s ‘COVIDSafe’, were deemed non-effective due to poor public uptake, stemming from issues with the technology (e.g., not functioning on Apple devices) [23]. Effectiveness plays a large role in the perceived technology benefits (e.g., privacy calculus), meaning our results must be considered a ‘best case scenario’ or benchmark for technology uptake in Taiwan.

### 4.3. Real World Applications

The ‘Taiwan Social Distancing App’ closely resembles our hypothetical Bluetooth scenario and Government app scenario with a local storage option. Our findings suggest that app acceptance would be improved by focusing policy and advertisement on the app’s public health benefits (to others, not to oneself), the app’s privacy measures, and by highlighting app acceptance and uptake in the community, thereby improving perceptions of public compliance. Focusing on the app’s privacy measures (e.g., a data deletion ‘sunset’ clause and its use of decentralized data storage) has the additional benefit of increasing technological transparency, a factor also associated with improved public health technology uptake [37]. Our recommendations may assist the Taiwanese Government in converting our observed high levels of app acceptance into actual app uptake—an issue that has affected countries such as Australia [23] and Germany [27], who also displayed high levels of app acceptance, yet experienced poor rates of uptake with their respective COVIDsafe and CORONA-WARN apps.

### 4.4. Comparison with Prior Work

This work extends previous descriptive studies on the attitudes towards tracing technologies among young Taiwanese adults [28] with representative national samples. We also elaborate on the descriptive accounts of studies in Germany [27], Australia [23], and the United Kingdom [25], with Bayesian regression modeling providing predictive factors for app uptake. Building upon this previous work, we see that education, privacy concerns, and personal benefits are key factors to app uptake (similar findings to the United Kingdom and Germany). We also see that uptake is dependent upon the perceived compliance of others (similar descriptive findings were observed in Australia for the COVIDSafe App). The comparative accounts of these studies combine to inform government decisions and policy making at both a national and international level.

## 5. Conclusions

With vaccine rollouts delayed, and new COVID-19 variants being discovered, the COVID-19 pandemic continues to pose a public health and economic threat to countries around the world. Through a series of representative Taiwanese surveys, our study found the acceptance of three tracing technologies—a Government app, telecommunication network tracking, and the Apple/Google Bluetooth exposure notification system—to be very high among the Taiwanese public. Our findings provide clear policy suggestions to assist the public uptake of Taiwan’s ‘Social Distancing App’. These results may also prove informative for other countries seeking technological solutions that allow economies to reopen without inadvertently reigniting the pandemic.

## Figures and Tables

**Figure 1 ijerph-19-03323-f001:**
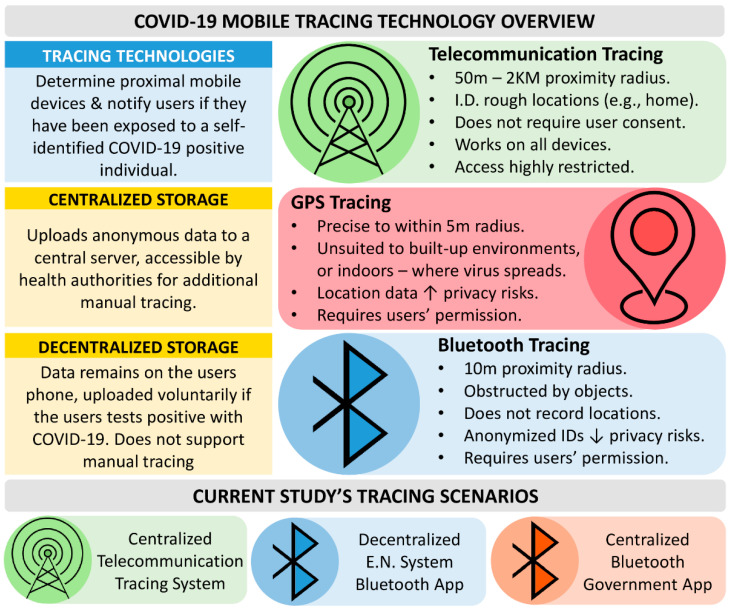
COVID-19 mobile tracing technologies, storage options, and the three tracing scenarios surveyed in the current study. A detailed description of each tracing scenario is presented in the Appendix A.

**Figure 2 ijerph-19-03323-f002:**
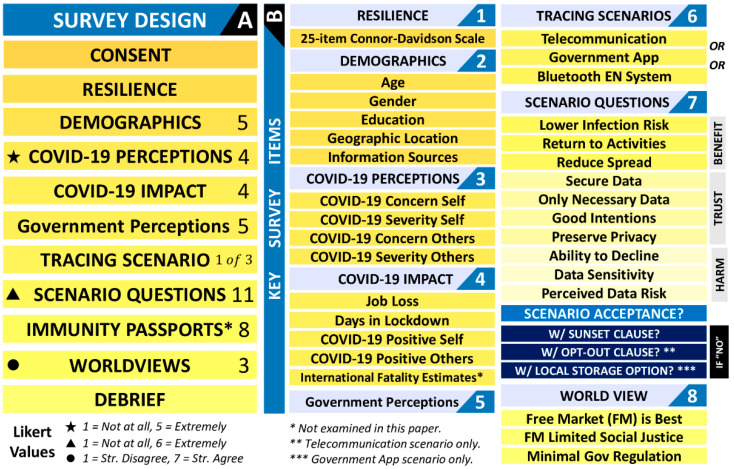
(**A**) The survey order, the number of items in each item-block (right), and associated Likert response options and values (bottom). (**B**) A break-down of the key items in this paper. Additional acceptance items are presented in navy blue for those participants who responded ‘no’ to the scenario acceptance item.

**Figure 3 ijerph-19-03323-f003:**
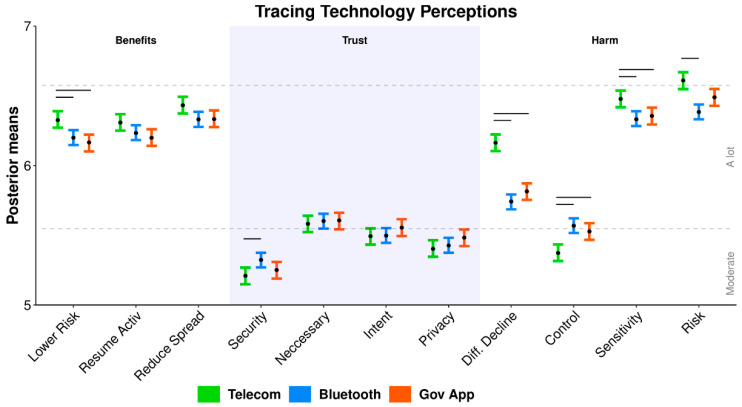
Ordinal regression mean posterior distributions for items assessing perceptions of each hypothetical tracing technology. Colored error bars display the 95% highest posterior density interval (HDI), black dots display the posterior mean, and black lines display where HDIs do not overlap within an item. Dotted lines depict boundaries separating the latent space into ordinal responses (1 = none to 6 = extremely; 4 = moderate).

**Figure 4 ijerph-19-03323-f004:**
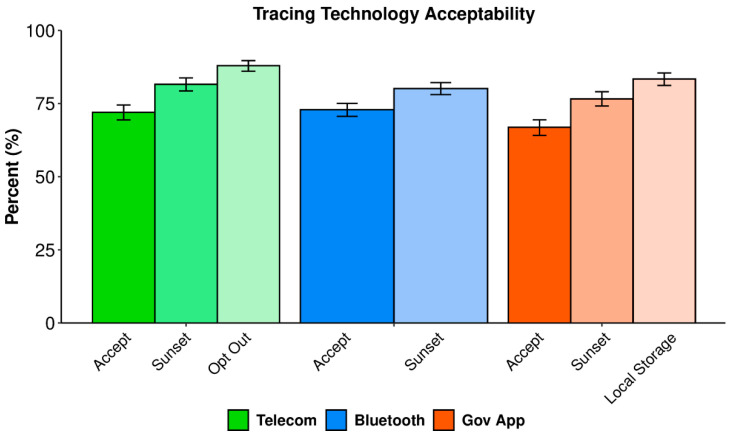
The acceptability and the conditional acceptability of each hypothetical tracing technology. Error bars are 95% Bayesian credible intervals. Highest density intervals within each technology do not overlap.

**Figure 5 ijerph-19-03323-f005:**
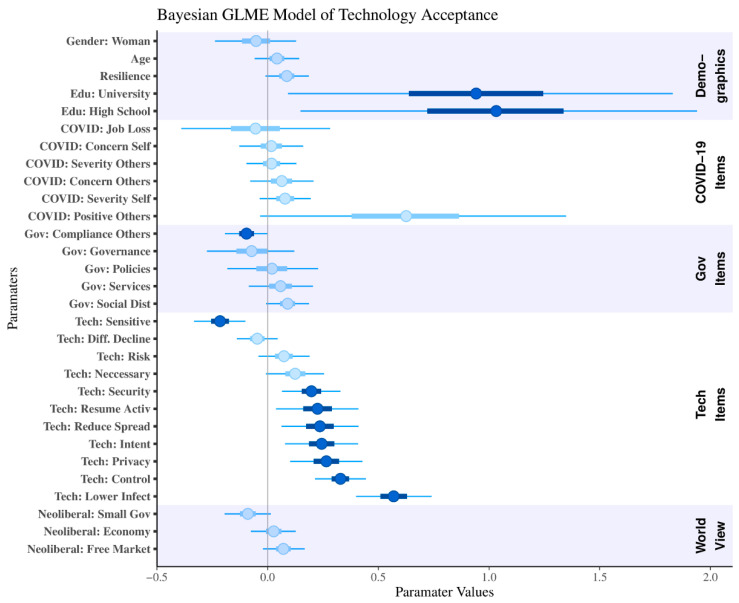
Bayesian generalized linear mixed effects model of tracing technology acceptance. Bars represent 50% of the parameter distribution centered on the parameter mean, tails display the 95% highest density interval. Opaque variables show instances where the posterior interval does not overlap zero.

**Table 1 ijerph-19-03323-t001:** Demographic information relevant to each sample. Edu: Education. Age is presented as the sample mean and standard deviation in years. Final N indicates the analyzed sample number after screening for the comprehension check item. Note: ‘Radio’ and ‘Other’ information source options are not displayed as they formed less than 1% of each sample, and percentages are rounded to the nearest integer if greater than 1.

		Sample 1	Sample 2	Sample 3	Sample 4
Sample	Initial *N*	1500	1500	1500	1500
	Final *N*	971	1018	939	897
Mean Age	Years (SD)	41 (12)	40 (12)	41 (12)	41 (12)
Gender (%)	Male	50	50	50	50
	Female	50	50	50	50
	Other Unspecified	0 0	0.2 0	0.13 0.07	0.07
Education (%)	Pre High School	1	1	1	1
	High School Grad University Grad	12 86	14 85	14 85	16
Info Source (%)	News Online/Print	59	63	63	64
	Television	26	23	22	19
	Social Media	12	11	13	16
	Friends and Family	2	2	1	1

**Table 2 ijerph-19-03323-t002:** COVID-19 perceived risk and impact, government perceptions, tracing technology (scenario) questions (benefits, trust and harm), and worldview items. Reverse-scored items are denoted with (R).

Item Block	Question	Label
Perception 1	How severe do you think novel coronavirus (COVID-19) will be for the general population?	Severity others
Perception 2	How harmful would it be for your health if you were to become infected COVID-19?	Severity self
Perception 3	How concerned are you that you might become infected with COVID-19?	Concern self
Perception 4	How concerned are you that somebody you know might become infected with COVID-19?	Concern others
Impact 1	Have you ever tested positive to COVID-19?	Positive self
Impact 2	Has somebody you know ever tested positive to COVID-19?	Positive others
Impact 3	How many days, if any, have you been in quarantine or self-isolation?	Lockdown days
Impact 4	Have you temporarily or permanently lost your job as a consequence of the COVID-19 pandemic?	Job loss
Gov 1	What percentage of the population do you think is complying with government policies regarding social distancing?	Social dist.
Gov 2	What percentage of the population do you think is complying with government policies regarding COVD-19?	Compliance
Gov 3	How satisfied are you with the current Government’s governance?	Governance
Gov 4	How satisfied are you with the current Government policies?	Policies
Gov 5	How satisfied are you with the current Government’s public services?	Services
Benefit 1	How confident are you that the described scenario would reduce your likelihood of contracting COVID-19?	Lower infection
Benefit 2	How confident are you that the described scenario would help you resume your normal activities more rapidly?	Resume activities
Benefit 3	How confident are you that the described scenario would reduce the spread of COVID-19?	Reduce spread
Trust 1	How secure are the data that would be collected?	Data security
Trust 2	To what extent is the Government (Apple/Google) only collecting the data necessary to achieve the purposes of the policy?	Data necessary
Trust 3	How much do you trust the Government (Apple/Google) to use the tracking data only to deal with the COVID-19 pandemic?	Trust intentions
Trust 4	How much do you trust the Government (Apple/Google) to be able to ensure the privacy of each individual?	Trust privacy
Harm 1	How difficult is it for people to decline participation?	Difficulty decline
Harm 2	To what extent do people have ongoing control of their data?	Ongoing control
Harm 3	How sensitive are the data being collected?	Data sensitivity
Harm 4	How serious is the risk of harm from the proposed scenario?	Risk
Worldview 1	Economic systems based on free markets unrestrained by government interference automatically work best to meet human needs.	Economy
Worldview 2	The free market system may be efficient for resource allocation, but it is limited in its capacity to promote social justice.	Free market (R)
Worldview 3	The government should interfere with the lives of citizens as little as possible.	Small gov

**Table 3 ijerph-19-03323-t003:** COVID-19 impact measures and mean psychological resilience, as measured by the CD-RISC scale, ranging from 0–100 with higher values indicating greater resilience.

	Sample 1	Sample 2	Sample 3	Sample 4
Mean Days in Lockdown (SD)	0.62 (3)	0.62 (3)	0.77 (3)	0.78 (3)
Job Loss (%)	7	9	8	8
Tested Positive Self (%)	0.53	0.73	0.8	0.33
Tested Positive Others (%)	3	2	3	3
Mean Resilience (SD)	64 (15)	64 (15)	65 (15)	65 (15)

## Data Availability

Not applicable.

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
