# Peer review of "High Acceptance of COVID-19 Tracing Technologies in Taiwan: A Nationally Representative Survey Analysis"

_ijerph, 2022, doi:10.3390/ijerph19063323_

Round 1

Reviewer 1 Report

Please find below the comment to improve your work.

-In general, this paper is mainly to measure acceptance for three hypothetical tracing technologies (telecommunication network tracing, a government App, and the Apple and Google Bluetooth exposure notification system) in four nationally representative Taiwanese samples.

- Line 172, Appendix A is mentioned, however, no Appendix exist in this paper. Please add the Appendix A.

- Line 287, in Figure 4, the number in percentage in each item should be indicated.

- Line 415, in clause 4.4, it is desirable to have a table to compare this result with previous work by identifying comparison criteria to show the uniqueness and originality of this work.

- In clause 1, introduction, it is necessary to have a paragraph describing the Contribution of this work.

Author Response

In general, this paper is mainly to measure acceptance for three hypothetical tracing technologies (telecommunication network tracing, a government App, and the Apple and Google Bluetooth exposure notification system) in four nationally representative Taiwanese samples.

- Line 172, Appendix A is mentioned, however, no Appendix exist in this paper. Please add the Appendix A.
Reply: Done. We have copied and pasted Appendix A & B to the main text of the revised manuscript.

- Line 287, in Figure 4, the number in percentage in each item should be indicated.
Reply: Done.

- Line 415, in clause 4.4, it is desirable to have a table to compare this result with previous work by identifying comparison criteria to show the uniqueness and originality of this work.
Reply: Done. This has been addressed in text as a table took more room than writing the comparisons directly. We have rewritten the comparison to prior work section to make this much clearer.

- In clause 1, introduction, it is necessary to have a paragraph describing the Contribution of this work.
Reply: Done. This has been added to the end of the introduction.

Reviewer 2 Report

Data have been collected almost 2 years ago, prior the rollout of the measures discussed in the paper.

Did the results presented in the paper correctly predicted real acceptance of the tracking technologies as they were applied?

Is there any correlation between the results presented and other COVID related issues like vaccination rejection etc?

Author Response

Data have been collected almost 2 years ago, prior the rollout of the measures discussed in the paper.

Did the results presented in the paper correctly predicted real acceptance of the tracking technologies as they were applied?

Reply: It is difficult to tell without a follow-up study of actual uptake that assess the same factors. There is no published data on the uptake of the Social Distancing App and app downloads have not been released by the Taiwan Center for Disease Control (TCDC) or AI Labs, the team responsible for making the app.

What we do know is that TCDC estimates suggest a 60% uptake was necessary for an effective app rollout in Taiwan. Estimates from Australia (Garrett et al., 2021a) and Germany (Kozyreva et al., 2021) – studies that used the same methodology as this paper – suggest App uptake is 18-20% lower than initial acceptance ratings.

Based on these findings and an initial acceptance rating of 73% in Taiwan, we would estimate App uptake to be around 53% – lower than the rate deemed necessary for the app to be most effective, but within a range that would assist in early COVID-19 isolation efforts.

We also know from modelling data in Germany of their CORONA-WARN app (Kozyreva et al., 2021), that education, privacy concerns, and perceived app effectiveness were predictive of real-world uptake. This finding is reflected in the current paper.

We now include these points in our discussion.

Is there any correlation between the results presented and other COVID related issues like vaccination rejection etc?

Reply: There may be some relationship, however, determining this empirically would require a separate investigation.

Nominal relationships to the uptake of immunity or vaccination passports can be seen (Garrett et al., 2021b) in that privacy concerns and education are key factors in their uptake as well.

We can also call upon the health belief model and a recent systematic review of the empirical data (Galanis et al., 2021) that shows vaccinations differ as a function of COVID-19 concern and perceived virus severity to one’s self; both of which are not factors that predict App uptake.

In short, it appears that the key difference in these measures is whether they benefit the individual directly (vaccination) or the community (Apps and immunity passports), and whether these health decisions incur a potential risk to one’s privacy.

Reviewer 3 Report

The paper is poorly written and organized, it needs complete rewriting 

Author Response

The paper is poorly written and organized, it needs complete rewriting

Reply: We suspect Reviewer 3’s primary issue was the similarity index, which we addressed in our initial statement to the editor. The paper has also been edited for clarity. We hope this addresses the concerns of Reviewer 3.

Please see below for our responses to the editor.

"With regards to the similarity index, it appears this score has been inflated and isn’t quite as problematic as it first appears. The similarity score includes 30 items which we detail below:

Items [1 – 3, 13] index the preprint of this manuscript stored in Minerva, accessible via PsyArXiv (https://psyarxiv.com/jg8rs/). This text it is identical and entirely our own work.

Items [4, 6, 7, 11, 18, 22, 29, 30] describe the survey items in Table 2, published in our other manuscripts (e.g., https://doi.org/10.1371/journal.pone.0244827, https://doi.org/10.1038/s41598-021-98249-5, and https://doi.org/10.1371/journal.pone.0245740).

Finally, items [5, 8 – 10, 15-16] are all standard text for publications in MDPI and Plos One (e.g., data availability statements) or otherwise constitute references. As such, these should not be counted towards our similarity index.

The remaining twelve items [12, 14, 17, 19-28] constitute less <1% similarity each."

Round 2

Reviewer 3 Report

The quality of this work is not worthy of impact factor publication, authors are advised to submit it at a conference for review.

Author Response

In our last revision, we have clearly stated the background and motivation of our study. The current study is an extension upon international work conducted in Australia, the United Kingdom, Germany, and on young adults in Taiwan, that have previously provided descriptive accounts of COVID-19 tracing app acceptance.

We have followed the MDPI editorial team's suggestions to revise our manuscript, including English language  and style.